# Plasma membrane overgrowth causes fibrotic collagen accumulation and immune activation in *Drosophila* adipocytes

Yiran Zang[1†], Ming Wan[1†], Min Liu[1], Hongmei Ke[1], Shuangchun Ma[1], Lu-Ping Liu[2,3], Jian-Quan Ni[2], José Carlos Pastor-Pareja[1]*

[1]School of Life Sciences, Tsinghua University, Beijing, China; [2]Gene Regulatory Lab, School of Medicine, Tsinghua University, Beijing, China; [3]Tsinghua Fly Center, School of Medicine, Tsinghua University, Beijing, China

**Abstract** Many chronic diseases are associated with fibrotic deposition of Collagen and other matrix proteins. Little is known about the factors that determine preferential onset of fibrosis in particular tissues. Here we show that plasma membrane (PM) overgrowth causes pericellular Collagen accumulation in *Drosophila* adipocytes. We found that loss of Dynamin and other endocytic components causes pericellular trapping of outgoing Collagen IV due to dramatic cortex expansion when endocytic removal of PM is prevented. Deposits also form in the absence of negative Toll immune regulator Cactus, excess PM being caused in this case by increased secretion. Finally, we show that trimeric Collagen accumulation, downstream of Toll or endocytic defects, activates a tissue damage response. Our work indicates that traffic imbalances and PM topology may contribute to fibrosis. It also places fibrotic deposits both downstream and upstream of immune signaling, consistent with the chronic character of fibrotic diseases.

*For correspondence: jose.pastor@biomed.tsinghua.edu.cn

†These authors contributed equally to this work

## Introduction

Basement membranes are polymers of extracellular matrix (ECM) proteins that underlie epithelia and surround organs in all animals (*Yurchenco, 2011*; *Kelley et al., 2014*). Their main constituent is *Collagen* IV, a helical trimer consisting of three α chains, capable of forming polymeric networks that interact with other ECM proteins. The fruitfly *Drosophila melanogaster* has emerged in recent years as an excellent model to study production of Collagen and other ECM proteins thanks to evolutionary conservation, powerful genetic tools and little genetic redundancy (*Denef et al., 2008*; *Martinek et al., 2008*; *Bunt et al., 2010*; *Haigo and Bilder, 2011*; *Drechsler et al., 2013*; *Lerner et al., 2013*; *Na et al., 2013*; *Devergne et al., 2014*; *McCall et al., 2014*; *Xiao et al., 2014*; *Zhang et al., 2014*). Two Collagen IV chains exist in *Drosophila*, encoded by *Collagen at 25C* (*Cg25C*, α1 chain) and *viking* (*vkg*, α2 chain). Apart from Collagen IV, BMs include Laminin, Nidogen and Perlecan, which are conserved from flies to humans as well (*Hynes, 2012*). In the larva, the main source of Collagen IV is the adipocytes of the fat body (*Pastor-Pareja and Xu, 2011*), known for their role in lipid storage and metabolic regulation, but also as an active secretory tissue which produces serum proteins and clotting factors normally present in the hemolymph (insect blood). The fat body is also a key effector of innate immunity, known to produce and secrete to the hemolymph large amounts of antimicrobial peptides in response to infections (*Lemaitre and Hoffmann, 2007*).

Secreted proteins like Collagen reach the extracellular space through a controlled series of membrane traffic events ensuring fusion of secretory vesicles with the plasma membrane (PM). Besides its role in cargo transport, membrane trafficking is a well-recognized driver of changes in cell shape and PM amount during morphogenesis (*Lecuit and Pilot, 2003*). Examples of morphogenetic

**eLife digest** In animals, so-called 'basement membranes' surround organs and help to both anchor certain tissue types together and control which molecules move between them. The basement membrane is made up of various proteins, and a large protein called Collagen IV is the most abundant component.

Collagen IV is made inside cells and packaged into bubble-like compartments called vesicles. These vesicles then merge with the cell membrane, which releases the collagen into the space outside the cell. Sometimes, after it has been released from the cell, Collagen IV forms harmful aggregates that the body finds difficult to break down. This condition is known as fibrosis, and can severely damage organs and tissues.

Zang, Wan et al. have now studied how fat cells—also known as adipocytes—in the fruit fly *Drosophila melanogaster* release Collagen IV. This fly is widely used to study collagen production because it is relatively easy to perform genetic investigations on it, and it releases collagen from its cells in the same way as many other species. Unexpectedly, it was observed that proteins that control a process known as endocytosis—which takes substances into the cell—are also involved in releasing Collagen IV from the cell. Zang, Wan et al. found that this is because endocytosis removes part of the cell membrane: if endocytosis is blocked, then the excess cell membrane traps Collagen IV molecules after they have been released, causing aggregates like those seen during fibrosis. However, artificially decreasing the amount of cell membrane restored normal collagen release.

Zang, Wan et al. next found that a pathway called Toll, which is important for protecting flies against infections, can also affect collagen release. When a protein that inactivates the Toll pathway is absent, too much cell membrane grows and Collagen IV forms aggregates as well. In both cases, Toll activation or lack of endocytosis, the aggregates trigger a reaction that damages the adipocytes. Understanding this reaction in more detail could help to develop treatments for conditions that produce fibrosis.

traffic in *Drosophila* include contraction of the amnioserosa during dorsal closure (*Mateus et al., 2011*) and widening of the lumen of tracheae (*Tsarouhas et al., 2007*). The best studied example of traffic-driven morphogenesis is perhaps blastoderm cellularization in the early *Drosophila* embryo. During blastoderm cellularization, fast directed PM growth results from membrane contributions from the secretory pathway (*Lecuit and Wieschaus, 2000*), endocytic membrane recycling (*Pelissier et al., 2003*; *Sokac and Wieschaus, 2008*; *Fabrowski et al., 2013*) and microvillar PM elaborations (*Figard et al., 2013*). While these examples highlight the potential of membrane traffic to elicit drastic changes in cell shape in the context of morphogenetic events, a role of in maintaining stable cortical morphology has not been addressed in detail and little is known on how cells normally regulate PM amount. Furthermore, the consequences for cell physiology of changes in this fundamental property are also unknown.

Handling of Collagen entails several challenges to secreting cells. Because of its large size, secretory transport of Collagen molecules requires carriers larger than regular COPII vesicles (*Saito et al., 2009*). Also, Collagen molecules undergo posttranslational modification along the secretory pathway by numerous Collagen-modifying enzymes such as glycosidases, and lysyl- and prolyl-hydroxylases, required for trimer formation (*Myllyharju and Kivirikko, 2004*). Prolyl-hydroxylation in particular is essential for trimer formation, mediated in *Drosophila* by the prolyl-4-hydroxylase PH4αEFB (*Pastor-Pareja and Xu, 2011*). Unlike fibrilar Collagen I, which flies lack, Collagen IV is secreted in functional form and does not require extracellular cleavage of the N- and C-terminal propeptides (*Khoshnoodi et al., 2008*). Therefore, and given its ability to form supramolecular assemblies, it is not known how Collagen IV avoids aggregation inside secreting cells or at their PM. Finally, because Collagen is resistant to most proteases, disposing of aggregates when they occur is highly problematic. Such is the case of fibrotic diseases, characterized by aberrant and excessive deposition of ECM due to persistent immune stimulation following diverse types of injuries (*Wynn and Ramalingam, 2012*). In a variety of tissues such as the liver, skin, kidney or fat tissue, fibrosis disrupts tissue organization and increases matrix stiffness, affecting normal cell and tissue physiology (*Hoffman et al., 2011*; *Friedman et al., 2013*).

In this work, we conducted a screening for genes involved in Collagen IV secretion by the fat body adipocytes in *Drosophila*. We found that loss of *shibire* and other endocytic genes leads to excess PM growth, which traps Collagen IV in the pericellular space. Pericellular Collagen IV trapping also results upon loss of loss of negative immune regulator *cactus*, which causes PM overgrowth through a Toll-dependent secretory burst. The accumulation of Collagen IV in the secreting adipocytes has two main consequences: Collagen IV deficit in the BMs of destination tissues and an immune response against adipocytes due to the abnormal ECM accumulation.

## Results

### Endocytic defects cause Collagen IV accumulation in *Drosophila* adipocytes

To gain new insights into Collagen biogenesis, we conducted a screening for genes affecting production of Collagen IV by fat body adipocytes, its main source in the *Drosophila* larva (*Figure 1A*). We used *BM-40-SPARC-GAL4* (*Venken et al., 2011*) to drive expression in adipocytes of the RNAi transgenes in the TRiP collection (8459 lines targeting 6200 genes) (*Ni et al., 2008*, *2011*) and analyzed the localization of Vkg-GFP, a functional GFP-trap fusion to the Collagen IV chain Vkg (*Morin et al., 2001*). While a majority of hits produced intracellular Collagen IV accumulation (full results to be published later), a distinct phenotypical category consisted of 60 genes causing accumulation at or near the PM (*Supplementary file 1*). Among the strongest hits in this category were two different RNAi transgenes targeting *shibire* (*shi*), encoding fly Dynamin, a GTPase involved in excision of endocytic vesicles (*Ferguson and De Camilli, 2012*). *shibire* knock-down (*shi^i*), as well as expression of dominant negative Dynamin (Shi^K44A) (*Moline et al., 1999*), caused Vkg accumulation in adipocytes (*Figure 1B*). In validation of the phenotype, antibody staining confirmed reduced Dynamin expression in *shi^i* cells, whereas the staining increased after Shi.K44A overexpression (*Figure 1C*), attesting to the sensitivity of the antibody. Since Collagen IV is a heterotrimer combining the α2 chain Vkg with Cg25C α1 chains, we performed a staining with an anti-Cg25C antibody we generated for this study (see *Figure 1—figure supplement 1*). This staining revealed that Cg25C, same as Vkg, accumulates in *shi^i* adipocytes (*Figure 1D*), thus confirming Collagen IV accumulation in these cells. The accumulation of Collagen IV occurs at the cell periphery, under a basement membrane surrounding the tissue in the wild type and which still forms in *shi^i* adipocytes (*Figure 1D*; see also Figure 3B later). Further validating the requirement of *shibire* in normal Collagen IV distribution, *shi^1* and *shi^2* thermosensitive mutations (*Kim and Wu, 1990*) also caused Vkg accumulation in adipocytes when larvae grew at restrictive temperature (*Figure 1—figure supplement 1*).

In addition to Dynamin, a number of hits showing a similar peripheral Collagen IV accumulation phenotype in the screening turned out to encode components of the endocytic machinery as well, such as Rab5 (*Figure 1E*), Clathrin heavy chain (*Figure 1F*), the Rab5-GAP RN-Tre, the AP-2 components AP-2α and AP-2μ and Hrs (*Figure 1—figure supplement 1*), suggesting that defective endocytosis was indeed the cause for the phenotype. Concurrent with the accumulation of Collagen IV in the adipocytes producing it, we found that the amount of Collagen IV present in destination BMs (*Figure 1G*) and in the hemolymph (*Figure 1—figure supplement 1*) was reduced in these conditions. In agreement with this, we observed elongation of the ventral nerve cord (*Figure 1H*), a deformation associated with Collagen IV reduction (*Pastor-Pareja and Xu, 2011*). We therefore conclude from these data that loss of *shibire* and other endocytic genes causes accumulation of Collagen IV in the adipocytes that normally secrete it and set out next to ascertain the mechanism by which this accumulation occurs.

### Collagen accumulation in endocytosis-defective cells is pericellular and autonomous

Accumulations of Collagen IV in *shi^i* adipocytes were found in close proximity to the PM, as revealed by the membrane marker myr-mRFP (myristoylation domain of Src fused to mRFP; *Figure 2A*) and by phalloidin staining of F-actin, normally enriched in the cell cortex (*Figure 2B*). To determine whether Collagen IV accumulations were intracellular or extracellular we stained *shi^i* adipocytes with the lipophilic, cell-impermeable dye FM4-64 (*Bolte et al., 2004*) to label membrane directly in contact with the extracellular space (PM). Short incubation with FM4-64 showed that the accumulations were surrounded by PM (*Figure 2C*) and, thus, likely extracellular. Labeling of membrane around the

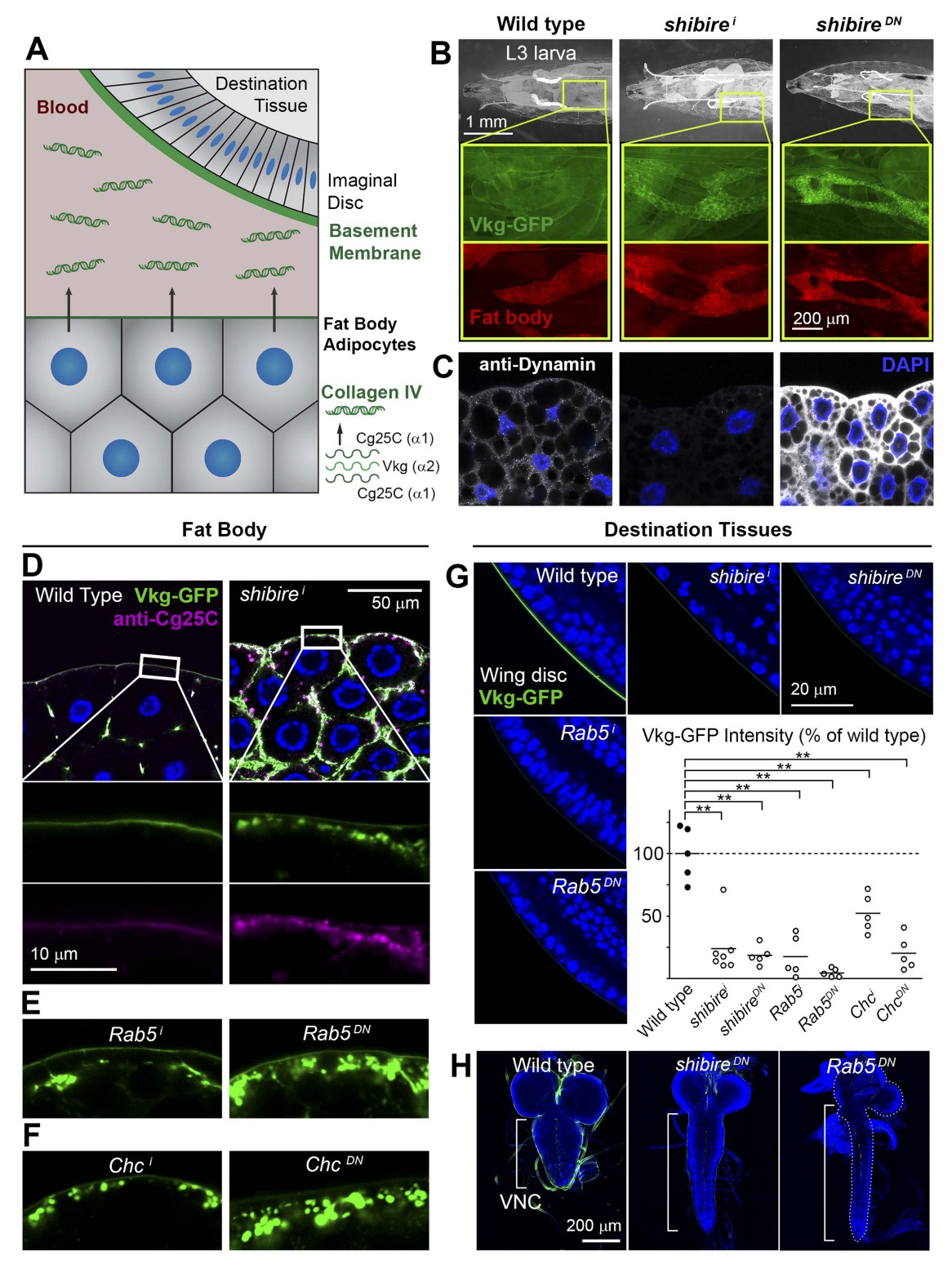

**Figure 1**. Endocytic defects cause Collagen accumulation in Drosophila adipocytes. (**A**) Schematic depiction of Collagen IV production, secretion and incorporation into basement membranes. (**B**) *shibire* knock-down (*BM-40-SPARC>shi^i*) and dominant negative *shibire^{K44A}* (*BM-40-SPARC>shi^{DN}*) cause Vkg-GFP accumulation in third instar larva adipocytes (marked with RFP). (**C**) Confocal images of third instar larva adipocytes stained with anti-Dynamin antibody. Staining is absent upon *shi* knock-down and increased by *shi^{K44A}* expression. Nuclei stained with DAPI. (**D**) Localization of Vkg-GFP and Cg25C (anti-Cg25C staining) in wild type and *BM-40-SPARC>shi^i* adipocytes. Collagen IV accumulates in the periphery of *shi^i* adipocytes. (**E**) Vkg accumulation in
*Figure 1. continued on next page*

*Figure 1. Continued*

*BM-40-SPARC>Rab5^i* and *>Rab5^{DN}* adipocytes. (**F**) Vkg accumulation in *BM-40-SPARC>Chc^i* and *>Chc^{DN}* adipocytes. (**G**) Presence of Vkg-GFP is reduced in discs from *BM-40-SPARC>shi^i*, *>shi^{DN}*, *>Rab5^i* and *>Rab5^{DN}* larvae. Vkg-GFP decrease quantified in graph. $n \geq 5$ for each genotype. Differences with wild type are in all cases significative (Mann–Whitney tests, **$p < 0.01$). (**H**) Elongation of the ventral nerve cord (VNC) in *BM-40-SPARC>shi^{DN}* and *>Rab5^{DN}*.

The following figure supplements are available for figure 1:

**Figure supplement 1**. (**A**) Confocal images of adipocytes from *shi^1 and shi^2* thermosensitive mutants.

**Figure supplement 2**. UAS-Dcr2 expression (*BM-40-SPARC-Gal4>UAS-Dcr2*) does not affect Collagen IV localization (Vkg-GFP) in adipocytes compared to Vkg-GFP control larvae (+) and larvae expressing GAL4 but not Dcr2 (*BM-40-SPARC-Gal4*).

accumulations was equally observed when *shi^i* adipocytes were fixed and stained with fluorescently-labeled fixable dextrans of molecular weights 70,000 (*Figure 2D*) and 3000 (*Figure 2—figure supplement 1*). To confirm that the Collagen IV accumulations in *shi^i* adipocytes were extracellular, we performed antibody stainings without permeabilizing the cells (no detergent in washing or blocking solutions). As a control, we stained *Tango1^i* adipocytes, known to retain Collagen IV intracellularly (*Pastor-Pareja and Xu, 2011*), and found that intracellularly retained Collagen IV could not be stained without permeabilization (*Figure 2E*). In contrast to this lack of staining, accumulations of Collagen IV in *shi^i* adipocytes were still labeled (*Figure 2E*), indicating that they were indeed extracellular. Altogether, these data show that Collagen IV accumulations in *shi^i* adipocytes are pericellular accumulations located outside of the PM (see also electron micrographs in *Figure 2—figure supplement 1*).

In light of the accumulation of Collagen IV, we first hypothesized that Collagen was internalized from the hemolymph (blood) by wild type adipocytes, which could explain pericellular accumulation when endocytosis was prevented. Contradicting this hypothesis, however, we found in mosaic experiments that accumulation of Vkg-GFP was suppressed when Vkg-GFP expression was knocked-down in the same cell (*Figure 2F*). This result demonstrates that Collagen IV accumulated in a Dynamin-deficient adipocyte originates autonomously in that same cell. Supporting the autonomous origin of accumulated Collagen IV, mosaic expression of Cg25C-RFP (see *Figure 2—figure supplement 1*) in *shi^i* adipocytes caused autonomous Collagen IV accumulation as well (*Figure 2G*). These experiments indicate that Collagen IV pericellularly accumulating in a *shi^i* adipocyte is not Collagen that failed to be endocytosed. Contrary to that, our data show that pericellularly accumulated Collagen IV, instead of ingressing into the cell, is outgoing Collagen secreted by that same cell and becoming extracellularly trapped in the cell cortex.

## Pericellular Collagen trapping in endocytosis-defective cells is due to PM overgrowth

In order to ascertain how endocytic defects cause pericellular trapping of outgoing Collagen, we proceeded to characterize the locus of its accumulation. The endocytic marker TfR (Transferrin Receptor, hTfR-GFP [*Henthorn et al., 2011*]) localized uniformly in the adipocyte PM and some intracellular vesicles (*Figure 2H*). Confirming that TfR is internalized and recycled back to the PM, knock-down of *Rab11*, required for recycling endosome trafficking (*Maxfield and McGraw, 2004*), caused TfR vesicles to fill the cytoplasm while no TfR was detected at the PM (*Figure 2J*). In Dynamin-deficient cells, in contrast, TfR concentrated heavily in deep PM pockets containing Collagen IV at their center (*Figure 2I*), suggesting that Collagen is pericellularly accumulated in PM pockets formed as a consequence of failed endocytosis.

To characterize in further detail the topology of the PM at the site of Collagen accumulation, we analyzed confocal and electron micrographs. Wild type adipocyte PM, which is flat during the first and second larval instars, becomes somewhat convoluted in the third instar (*Figure 3—figure supplement 1*), displaying multiple invaginations which often encircle surface lipid droplets (*Diaconeasa et al., 2013*). Compared to this, examination of endocytosis-deficient adipocytes revealed a striking increase in PM amount with respect to the wild type (*Figure 3A,B*). Quantification in both confocal and electron

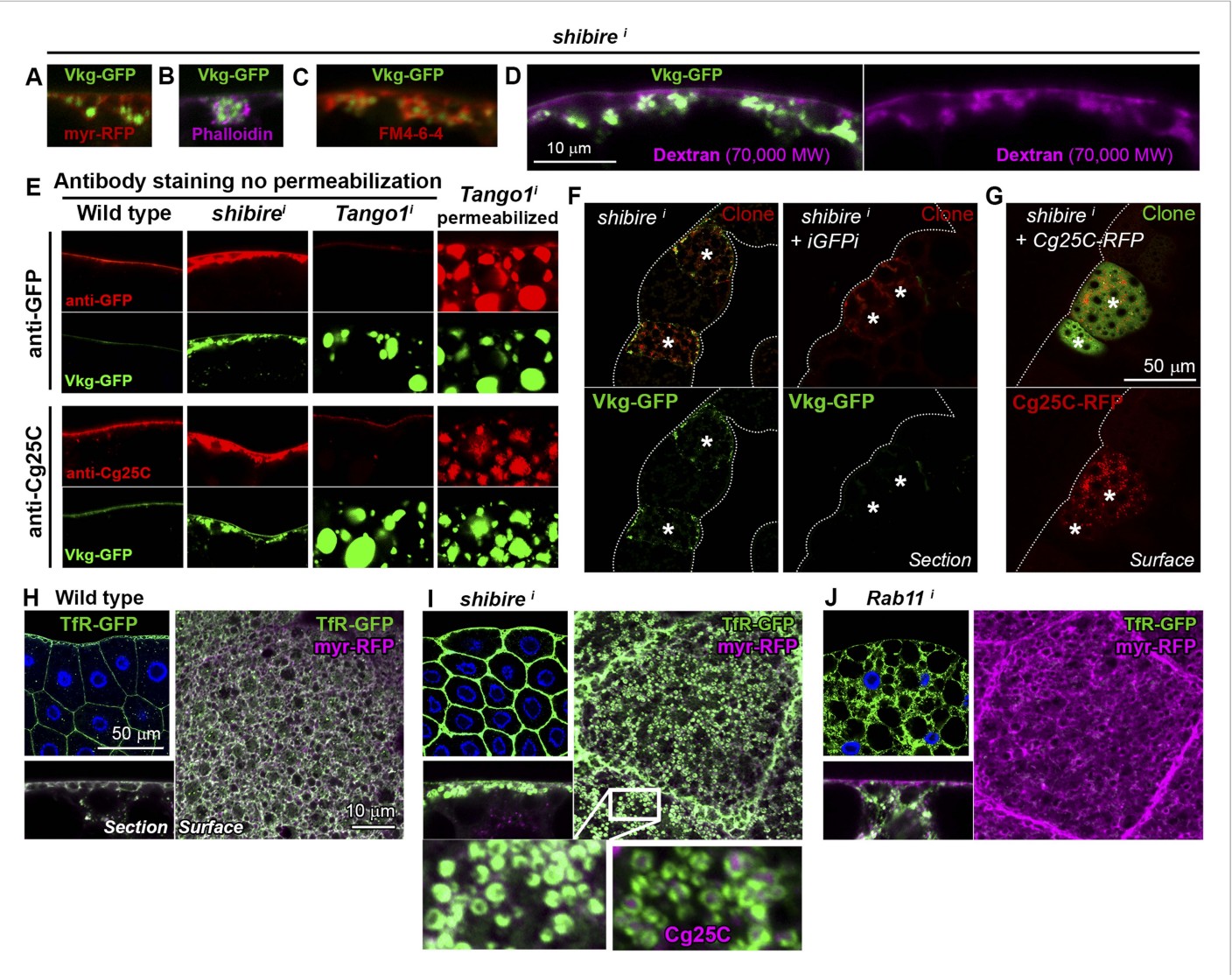

**Figure 2**. Collagen accumulation in endocytosis-defective cells is pericellular and autonomous. (**A**) Vkg-GFP accumulation in a *shi^i* adipocyte (*BM-40-SPARC>shi^i*) expressing membrane marker myr-RFP. (**B**) *shi^i* adipocyte stained with phalloidin (F-actin). (**C**) *shi^i* adipocyte stained with cell-impermeable membrane dye FM4-64, labelling plasma membrane (PM) around accumulations. (**D**) *shi^i* adipocyte stained with fixable Texas-Red-coupled Dextran (70,000 MW), labelling PM around accumulations. (**E**) Antibody stainings of wild type and *shi^i* adipocytes performed without permeabilization (no detergent) in order to detect extracellular Collagen IV. In contrast to the accumulations in *shi^i* adipocytes, intracellular accumulations of Collagen IV in *Tango1^i* adipocytes cannot be stained in the absence of permeabilization and are shown as a control. (**F**) Mosaic fat body (*act-GAL4>shi^i* flip-out clones, marked with RFP) showing Vkg-GFP accumulation in *shi^i* cells. Accumulation is suppressed by a GFP-targeting dsRNA (iGFPi). (**G**) Pericellular accumulation in mosaic *shi^i* fat body expressing Cg25C-RFP. Clones marked with GFP. (**H**) Localization of the endocytic marker Transferrin Receptor (*Cg>TfR-GFP*) in wild type adipocytes. (**I**) In *shi^i* adipocytes, TfR concentrates in PM pockets containing Collagen IV (anti-Cg25C). (**J**) In *Rab11^i* adipocytes, TfR localizes to intracellular vesicles that completely fill the cytoplasm. No TfR is detected at the PM.

The following figure supplement is available for figure 2:

**Figure supplement 1**. (**A**) Confocal images showing the PM of *shi^i* and *Tl^{10B}* adipocytes stained with fixable Texas-Red-coupled Dextrans (3000 and 70,000 MW), labelling PM around Collagen IV (Vkg-GFP) accumulations.

micrographs showed an increase in the depth of PM ingression into the cytoplasm and higher PM sinuosity in endocytosis-defective cells (*Figure 3C*; *Figure 3—figure supplement 2*). The opposite phenotype, PM flattening, along with a vesicle-filled cytoplasm, resulted from *Rab11* knock down. These results indicate that both endocytosis and membrane recycling are critical to maintain normal

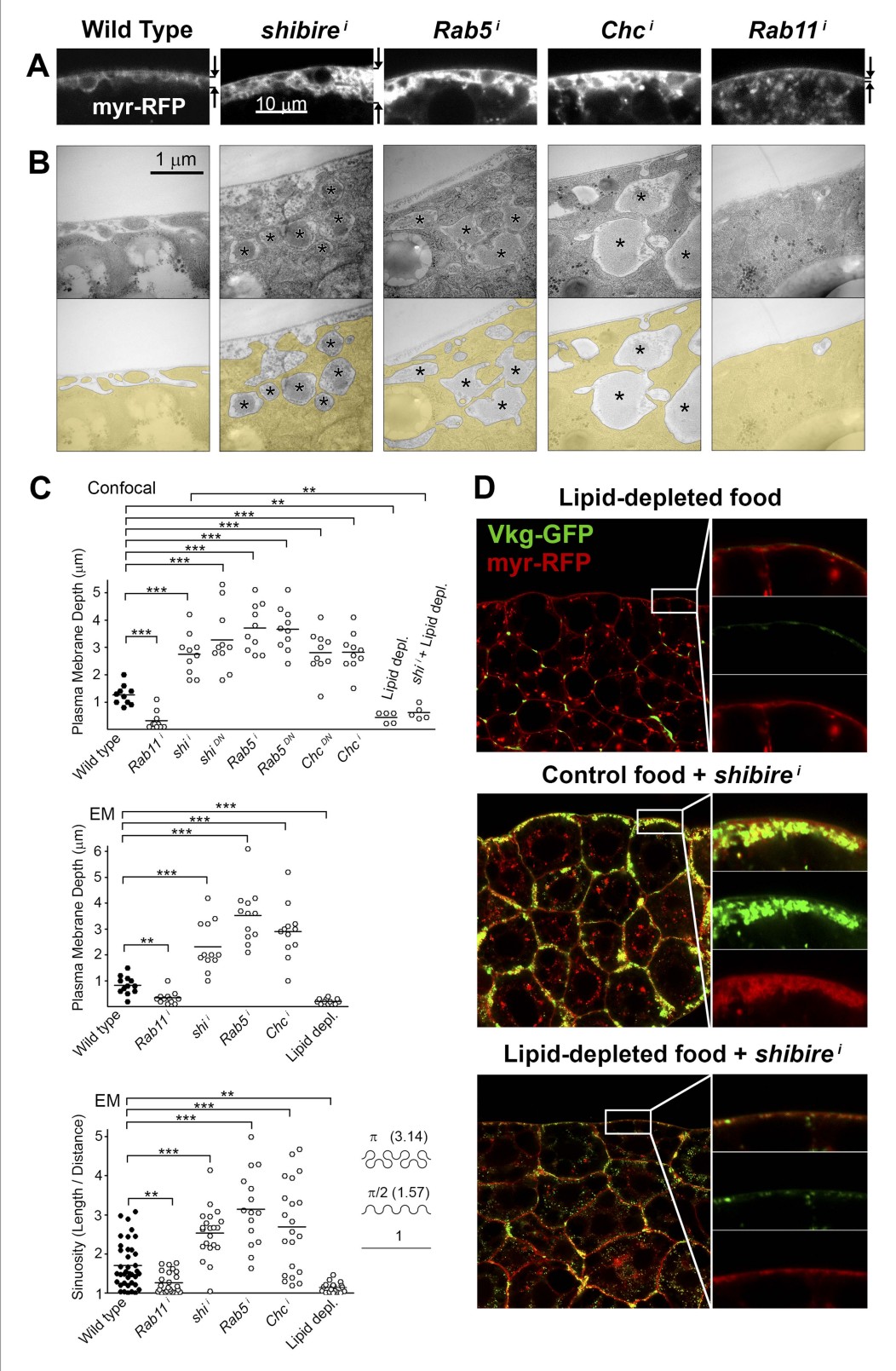

**Figure 3.** Pericellular Collagen trapping is due to PM overgrowth. (**A**) Confocal sections of adipocyte PM (myr-RFP marker). PM expansion is observed in *BM-40-SPARC>shi^i*, *>Rab5^i* and *>Chc^i* adipocytes, whereas PM flattening occurs in *>Rab11^i* adipocytes along with accumulation of intracellular vesicles. (**B**) Electron micrographs of adipocyte

*Figure 3. continued on next page*

*Figure 3. Continued*

PM from control, *BM-40-SPARC>shi$^i$*, *>Rab5$^i$*, *>Chc$^i$* and *>Rab11$^i$* larvae. Internal cell volume indicated through yellow transparency. Asterisks mark pericellular deposits. (**C**) Quantification of PM depth and sinuosity (see *Figure 3—figure supplement 2*). Depth measurements obtained from confocal (n = 12) and electron (n = 10) micrographs. PM sinuosity is the ratio between the length of PM between two points on that membrane and the linear distance separating them (n ≥ 15). Differences with controls were significative as indicated (Mann–Whitney tests, \*\*p < 0.01, \*\*\*p < 0.001). (**D**) Adipocytes from wild type larvae grown on lipid-depleted food, *BM-40-SPARC>shi$^i$* larvae and *BM-40-SPARC>shi$^i$* larvae grown on lipid-depleted food. Vkg-GFP accumulation and PM excess are both suppressed by lipid-depletion.

The following figure supplements are available for figure 3:

**Figure supplement 1**. (**A**) Confocal images of fat body dissected from first, second and third instar larvae.

**Figure supplement 2**. Schematic explanation of PM sinuosity and depth measurements performed in electron micrographs.

PM amount and cortical morphology, suggesting that Collagen IV is trapped in an abnormally expanded cell cortex when endocytic removal of membrane from the PM is prevented.

To confirm that an excessive amount of PM can act as a barrier to Collagen release after secretion, we aimed at artificially decreasing PM amount in adipocytes. To that end, we cultured flies in medium depleted of lipids through chloroform extraction (*Palm et al., 2012*). In larvae thus grown, we found that the amount of adipocyte PM was greatly reduced (*Figure 3C,D*; *Figure 3—figure supplement 1*). Furthermore, lipid-depleted food suppressed both PM excess and pericellular Collagen accumulation in Dynamin-deficient adipocytes (*Figure 3C,D*), indicating that PM overgrowth was indeed responsible for pericellular Collagen trapping. These data, in all, show that PM excess resulting from lack of endocytic membrane removal leads to a hyperconvoluted PM, which in turn causes pericellular trapping of Collagen IV.

## Pericellular deposits in adipocytes are fibrotic

We asked next whether other proteins besides Collagen IV were pericellularly trapped due to PM overgrowth. Apart from Collagen IV, the main components of basement membranes are Perlecan, Laminin and Nidogen (*Yurchenco, 2011*). Whereas evidence exists of significant Laminin and Nidogen production outside the fat body (*Urbano et al., 2009*; *Zhu et al., 2012*), production of Perlecan has not been studied. Through iYFPi (in vivo YFP interference, *Figure 4A*), we knocked down expression of Perlecan-YFP (*trol$^{CPTI-002049}$* [*Rees et al., 2011*]), a YFP-trap insertion predicted to label all Perlecan isoforms and found that Perlecan present in imaginal discs originated entirely in the fat body (*Figure 4B*), same as Collagen IV. Also similar to Collagen IV, Trol-YFP was pericellularly accumulated in endocytosis-defective adipocytes (*Figure 5A,B*). This accumulation of Trol (*terribly reduced optic lobes*) depended on Collagen IV, as it was suppressed by Collagen IV knock-down (*Figure 5C*). Through antibody staining, we confirmed Perlecan accumulation (*Figure 5—figure supplement 1*) and additionally observed accumulation of Nidogen, again in a Collagen-dependent manner (*Figure 5D*), but not Laminin (anti-LanB1 staining, not shown).

The fact that the aggregates contained basement membrane components Collagen IV, Perlecan and Nidogen suggested that they were in essence fibrotic ECM deposits. Indeed, Vkg accumulation was suppressed by knock down of the Prolyl-4-hydroxylase PH4αEFB (*Figure 5E*), in the absence of which Collagen IV is still secreted to the blood as non-functional monomeric chains (*Pastor-Pareja and Xu, 2011*). Additionally supporting the fibrotic nature of the deposits, the secretion marker secr-GFP (GFP coupled to a signal peptide [*Pfeiffer et al., 2002*]) did not accumulate in the expanded cortex of *shi$^i$* adipocytes (*Figure 5F*). We also examined two other non-ECM proteins secreted by the fat body, 26-29-protease and Ferritin 1HCH, and none of them accumulated pericellularly upon *shi* knock down (*Figure 5—figure supplement 1*). These results demonstrate that not all proteins secreted by the fat body are pericellularly trapped as a result of PM overgrowth. On the contrary, our data support the notion that the aggregates consist of ECM proteins like trimeric Collagen IV, Perlecan

and Nidogen, and thus are fibrotic deposits. Furthermore, our results suggest that Collagen IV is the main protein in this deposits, as accumulation of Perlecan and Nidogen depended on the presence of Collagen IV.

## Fibrotic deposits and PM overgrowth upon Toll activation

Having established that extracellular deposits caused by PM overgrowth were not general protein aggregates but rather fibrotic ECM aggregates, we decided to investigate their wider biological effects. To do that, we first turned our attention to *cactus*, another PM accumulation hit in our screening (*Figure 6A*). *cactus* (*cact*) encodes a negative regulator of the Toll signaling pathway, a key mediator of innate immunity (*Roth et al., 1991*). We confirmed activation of Toll signaling in *cact*[i] adipocytes (Dorsal nuclear accumulation) and further validated the phenotype by observing pericellular Collagen accumulation in *cact*[4] mutants (*Roth et al., 1991*) (*Figure 6—figure supplement 1*). Expression of the constitutively active mutant receptor Toll[10B] also produced pericellular Collagen accumulation (*Figure 6B*), as did infection with Toll-activating Gram+ bacteria *Micrococcus luteus* (*Figure 6C*; *Figure 6—figure supplement 1*). Examination of *cact*[i] and Toll[10B] adipocytes in confocal and electron micrographs revealed PM overgrowth (*Figure 6D,E*), similar to *shi*[i] adipocytes. Also similar to *shi*[i] adipocytes, PM overgrowth upon Toll activation was suppressed by lipid-depleted food (graph in *Figure 6E*). Moreover, pericellular Collagen IV deposits were extracellular (antibody staining without permeabilization, *Figure 6—figure supplement 1*), disappeared upon *PH4αEFB* knock-down (*Figure 6F*) and did not contain secr-GFP (*Figure 6G*), but contained Perlecan (*Figure 6H*), indicating accumulation of fibrotic material, same as in endocytosis-defective cells.

Despite these similarities with endocytosis-defective cells, we could not find evidence of endocytosis/recycling defects when we examined TfR-GFP localization in *cact*[i] and Toll[10B] adipocytes (*Figure 6—figure supplement 1*), suggesting that PM overgrowth upon Toll activation did not stem from reduced endocytosis. Toll activation, on the other hand, is known to potently induce production of secreted antimicrobial peptides like Drosomycin, Defensin and Metchnikowin (*Lemaitre et al., 1996*), which led as to consider increased secretion of these Toll targets as an alternative explanation to PM overgrowth upon Toll activation. Consistent with this, knock-down of Dif, a transcription factor involved in expression of Toll target genes in the adult (*Rutschmann et al., 2000*) and larval fat body (see *Figure 6—figure supplement 1*), suppressed PM overgrowth (graph in *Figure 6E*) and pericellular Vkg-GFP trapping (not shown), suggesting that the transcriptional response to Toll activation contributed to Collagen IV accumulation in *cact*[i] and Toll[10B] adipocytes. Also consistent with a burst in secretory activity causing the PM overgrowth, the cytoplasm of *cact*[i], Toll[10B] and *M. luteus*-infected adipocytes was filled with Drosomycin vesicles or granules (*Figure 6I*; *Figure 6—figure supplement 1*). Furthermore, knock-down of *Rab1*, required for ER-to-Golgi transport in the secretory

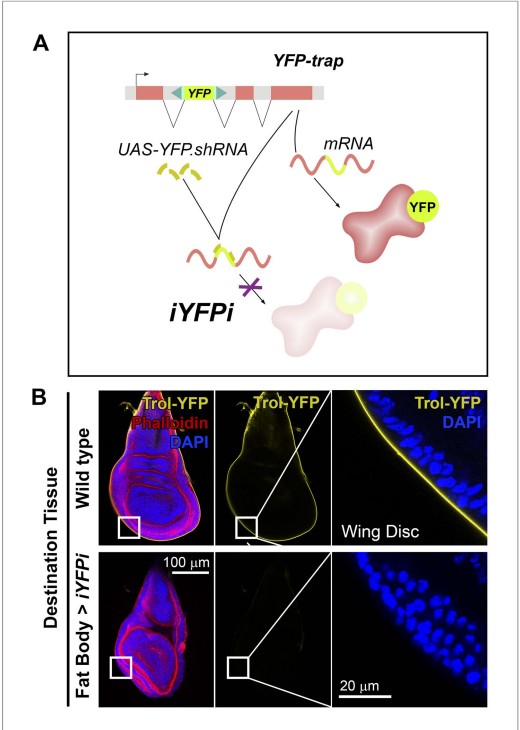

**Figure 4**. Perlecan, like Collagen IV, originates in the fat body. (**A**) Schematic representation of the in vivo YFP interference strategy (iYFPi) to knock-down expression of YFP-trapped Perlecan (Trol-YFP) and ascertain its tissue of origin. Expression of a short hairpin RNA targets the YFP sequence in the YFP-trapped mRNA for degradation through RNAi. (**B**) Localization of Perlecan (Trol-YFP trap) in wing discs from *trol*[CPTI-002049]/Y flies. iYFPi in the fat body (*BM-40-SPARC>iYFPi*) eliminates expression of Trol-YFP in the wing disc and produces tissue hyperconstriction, a previously described *trol* loss-of-function phenotype (*Pastor-Pareja and Xu, 2011*). Phalloidin staining of F-actin in red to reveal disc deformation.

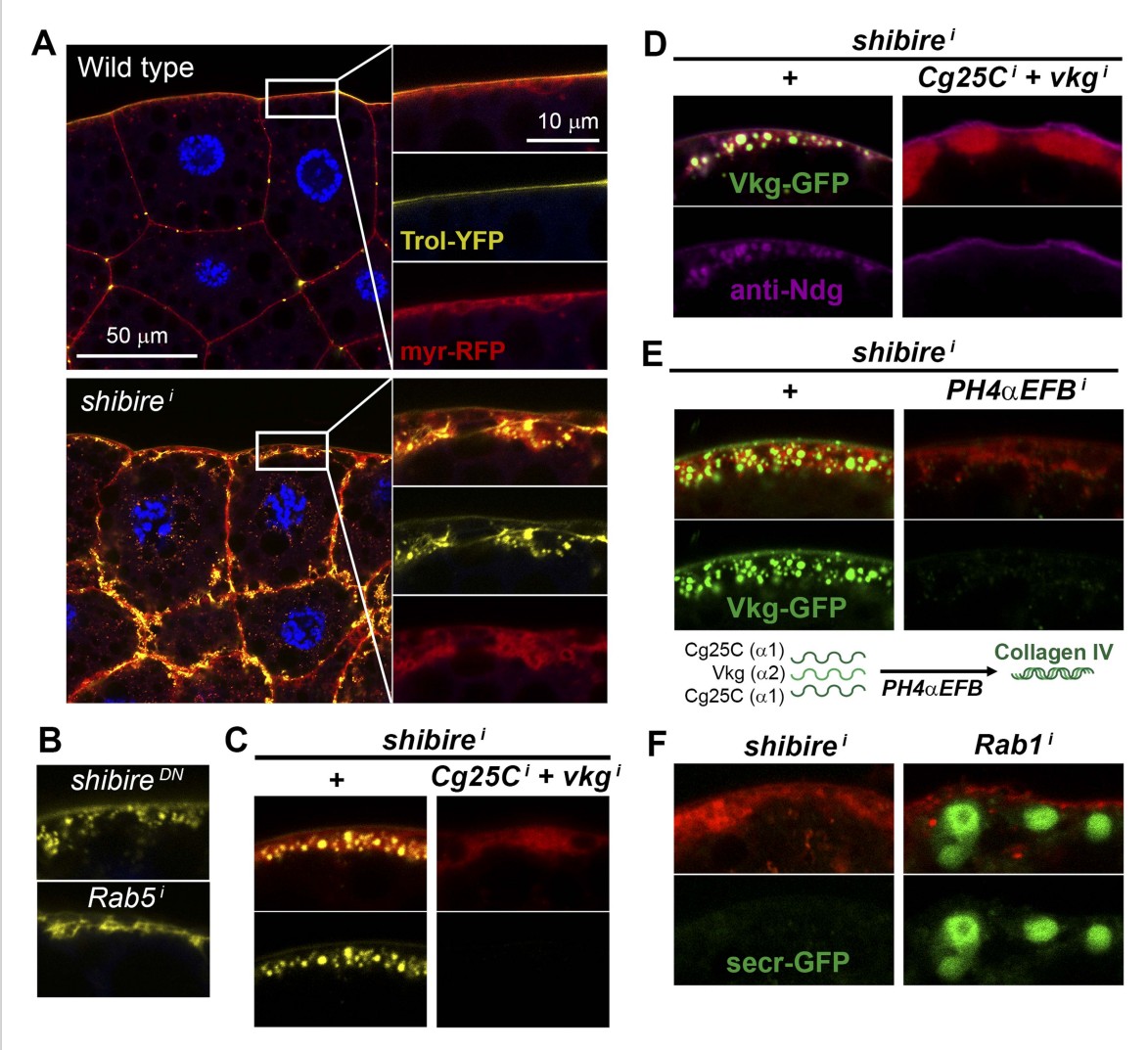

**Figure 5**. Pericellular deposits in adipocytes are fibrotic. (**A**) Localization in wild type and *BM-40-SPARC>shi^i* adipocytes of Trol-YFP (*trol^CPTI-002049*; see *Figure 4*). Perlecan accumulates in *>shi^i* adipocytes. (**B**) Perlecan accumulation in *BM-40-SPARC>shi^DN* and *>Rab5^i* adipocytes. (**C**) Perlecan accumulation in *r4>shi^i* adipocytes is suppressed by Collagen IV knock down. (**D**) Pericellular Nidogen accumulation (anti-Ndg staining) in *r4>shi^i* adipocytes is suppressed by Collagen IV knock down. (**E**) Pericellular Vkg accumulation in *BM-40-SPARC>shi^i* adipocytes is suppressed by knocking down prolyl-hydroxylase PH4αEFB. (**F**) *BM-40-SPARC>shi^i* adipocytes do not accumulate secretion marker secr-GFP. Intracellular secr-GFP retention in *BM-40-SPARC>Rab1^i* is shown as a control.

The following figure supplement is available for figure 5:

**Figure supplement 1**. (**A**) PM of fat body adipocytes stained with anti-Trol antibody (magenta).

pathway, caused not just abundant intracellular Drs-GFP retention in *Toll^10B* adipocytes, but also suppression of PM excess (*Figure 6J*), implicating membrane input from the secretory pathway in PM overgrowth. In all, our analysis indicates that PM overgrowth and fibrotic deposits upon Toll activation result from increased secretory activity.

## Immune response to fibrotic deposits

Our experiments indicated that fibrotic aggregates could form downstream of Toll immune signaling. We therefore decided to characterize further the immune effects of these deposits. In our screening, 32 out of 70 hits producing pericellular Collagen IV accumulation showed clear signs of a fat body

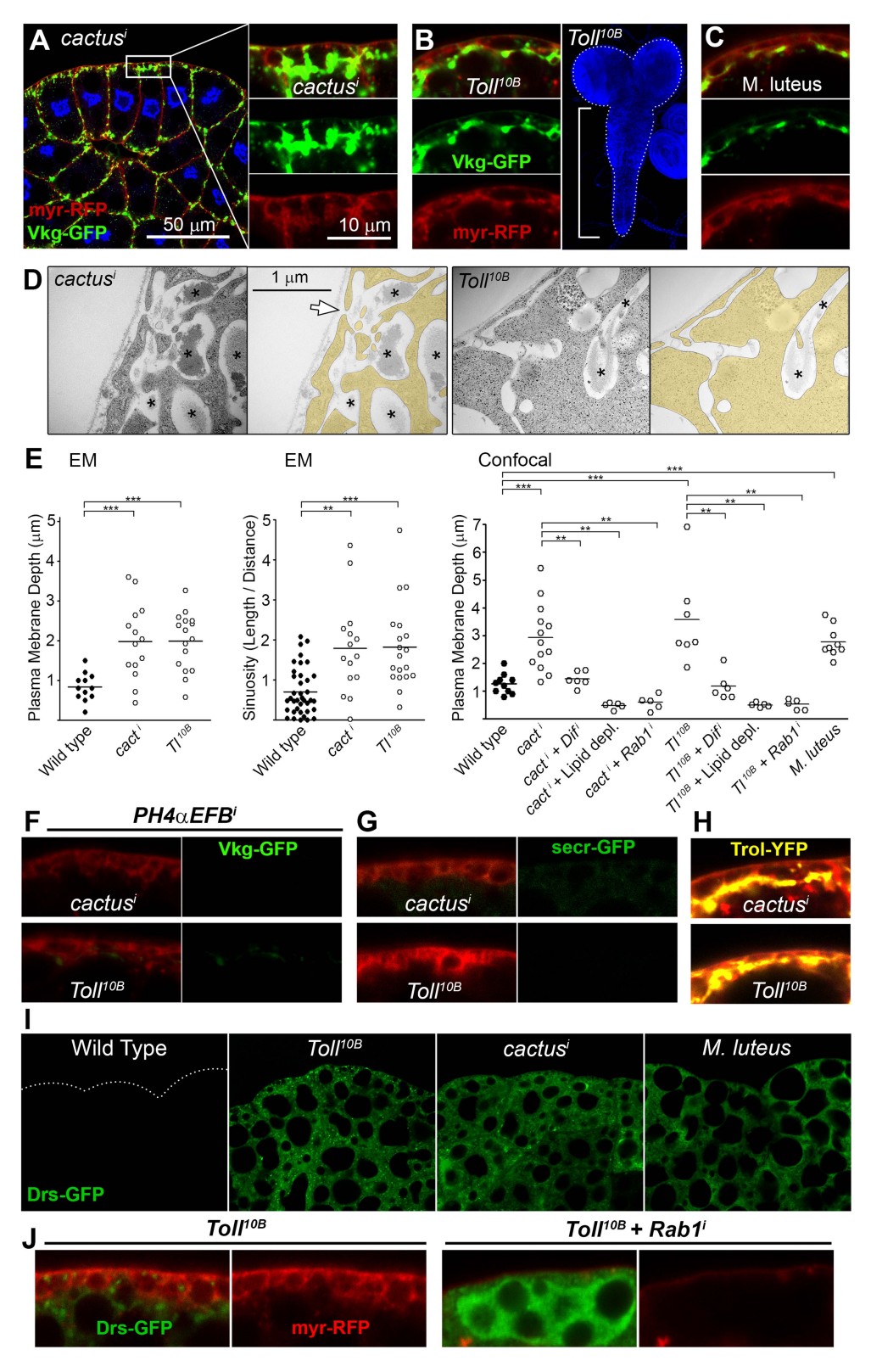

**Figure 6**. Fibrotic deposits and PM overgrowth upon Toll activation. (**A**) Pericellular Vkg deposits (Vkg-GFP) and PM overgrowth in *BM-40-SPARC>cact$^i$* adipocytes. (**B**) Vkg deposits and VNC elongation in *BM-40-SPARC>Tl$^{10B}$* larvae. (**C**) Vkg deposits and PM overgrowth in adipocytes 1 day after infection with *Micrococcus luteus*. (**D**) Electron
*Figure 6. continued on next page*

*Figure 6. Continued*

micrographs of *BM-40-SPARC>cact^i* and *>Tl^{10B}* adipocytes. The arrow marks connection of the deposits to the extracellular space. (**E**) Measurements of PM depth and sinuosity in adipocytes of indicated genotypes. Depth measurements obtained from confocal (n ≥ 7) and electron micrographs (n ≥ 10). Sinuosity measured in electron micrographs (n ≥ 15). Differences with wild type or appropriate control as indicated were significative in all cases (Mann–Whitney tests, **p < 0.01, **p < 0.001). (**F**) Pericellular Vkg accumulation in *BM-40-SPARC>cact^i* and *>Tl^{10B}* adipocytes is suppressed by knocking down prolyl-hydroxylase PH4αEFB. (**G**) Secretion marker secr-GFP does not accumulate in *BM-40-SPARC>cact^i* or *>Tl^{10B}* adipocytes. (**H**) Pericellular Perlecan deposits (Trol-YFP) in *BM-40-SPARC>cact^i* and *>Tl^{10B}* adipocytes. (**I**) Induction of antimicrobial peptide Drosomycin (Drs-GFP) fills adipocyte cytoplasm in *BM-40-SPARC>cact^i*, *>Tl^{10B}* and *Micrococcus luteus*-infected larvae. (**J**) Rab1 knock-down causes intracellular Drosomycin retention and suppresses PM overgrowth in *BM-40-SPARC>Tl^{10B}* adipocytes.

The following figure supplement is available for figure 6:

**Figure supplement 1**. (**A**) Nuclear accumulation of the Toll downstream transcription factor Dorsal (anti-Dorsal staining) in *BM-40-SPARC>cacti* adipocytes.

---

melanization response, including *cact*, *shi*, *Rab5* and *Chc* (**Supplementary file 1**). Melanization is an insect immune response characterized by blackening of the affected tissue, usually accompanied by hemocyte (blood cell) recruitment (**Minakhina and Steward, 2006**). Because *BM-40-SPARC-GAL4* and *Cg-GAL4*, the strong fat body drivers we had used so far, are also expressed in blood cells, we tested weaker fat body drivers *ppl-GAL4* and *r4-GAL4*, inactive in blood cells, and found that melanization still occurred (**Figure 7A,B**), ruling out that interfering with blood cell function caused the response. Furthermore, confirming the involvement of Collagen deposits in the response, both the number of *r4>shi^i* larvae displaying melanization and the extent of it decreased when we additionally knocked down Collagen IV or PH4αEFB (**Figure 7C,D**). Reduction of Collagen IV and PH4αEFB also reduced melanization of *r4>Toll^{10B}* larvae (**Figure 7D**) and completely rescued their pupal lethality. These results show that Collagen deposits either trigger or significantly contribute to fat body melanization in these conditions.

Given that Toll activation is long known to cause melanization (**Gerttula et al., 1988**), we examined Toll activity in *shi^i* adipocytes, but found that *shibire* knock-down did not cause Toll pathway activation, as evidenced by lack of Drs-GFP induction and absence of nuclear Dorsal localization (not shown). In contrast, both *shibire* knockdown and Toll activation caused activation of the c-Jun N-terminal kinase (JNK) and Janus kinase/Signal Transducer and Activator of Transcription (JAK/STAT) pathways, known to act together in response to tissue damage in other fly tissues (**Pastor-Pareja et al., 2008**; **Buchon et al., 2009**). JNK targets *puckered* (*puc*) and *Matrix metallo-protease 1* (*Mmp1*) (**Uhlirova and Bohmann, 2006**) were induced in *shi^i*, *Toll^{10B}* and *cact^i* adipocytes (**Figure 7E–G**). As for JAK/STAT signaling, we observed induction of the JAK/STAT reporter 10XSTAT-GFP (**Ekas et al., 2006**) (**Figure 7G**) and highly upregulated expression of unpaired 3 (*upd3*), encoding a JAK/STAT-activating cytokine (**Figure 7H**). From these data, we conclude that fibrotic deposits caused by endocytic defects or Toll provoke tissue damage in adipocytes and stimulate an innate immune response, as evidenced by tissue melanization, blood cell recruitment, activation of JAK/STAT signaling and JNK (**Figure 8**).

## Discussion

### Excess PM as a barrier to Collagen release after secretion

In this study, we investigated the release of Collagen IV by the fat body adipocytes in *Drosophila* larvae. Through a genetic screening, we discovered that reducing endocytosis caused unexpected Collagen IV accumulation in secreting cells (**Figure 1**). In investigating the cause for such a paradoxical phenotype, our experiments revealed that Collagen IV was effectively secreted and reached the extracellular space, but it was then trapped in fibrotic deposits at the cell periphery (**Figure 2**). Further experiments led us to conclude that endocytic membrane removal and recycling are critical to maintain normal PM amount and that excess of PM when endocytic membrane removal was prevented resulted in a hyperconvoluted cell cortex where Collagen IV became entrapped (**Figure 3**).

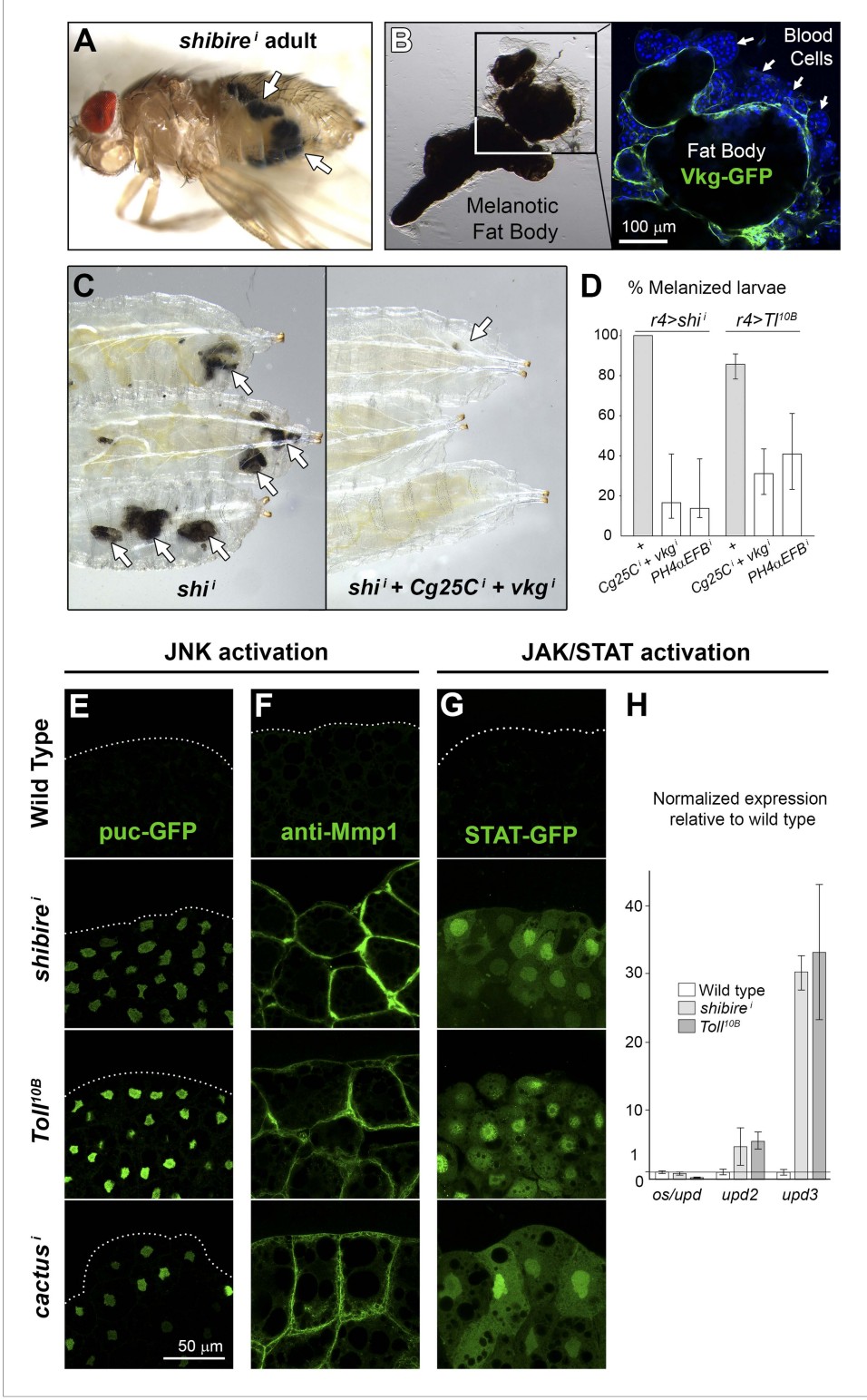

**Figure 7**. Immune response to fibrotic deposits. (**A**) Melanotic fat body in a *ppl>shi^i* fly. (**B**) Melanized fat body from an *r4>shi^i* larva. Hemocytes (blood cells) encapsulate the tissue. (**C**) Knock-down of Collagen IV reduces fat body melanization in *r4>shi^i* larvae. Cultures maintained at 30˚C. (**D**) Percentage of larvae displaying signs of melanization in indicated genotypes. n ≥ 30 per genotype. Differences with *r4>shi^i* and *>Tl^{10B}* controls were significative ($\chi^2$ tests, ***$p < 0.001$). Cultures maintained at 30˚C. (**E**) Induction of c-Jun N-terminal kinase (JNK) downstream *puckered* (puc-GFP enhancer trap) in *BM-40-SPARC>shi^i*, *>Tl^{10B}* and *>cact^i* adipocytes. (**F**) Induction of Matrix

*Figure 7. Continued*

Metallo-Protease 1 (anti-Mmp1 staining) in *BM-40-SPARC>shi[i]*, *>Tl[10B]* and *>cact[i]* adipocytes. (**G**) Expression of JAK/STAT activity reporter 10XSTAT-GFP in *BM-40-SPARC>shi[i]*, *>Tl[10B]* and *>cact[i]* adipocytes. (**H**) Expression of JAK/STAT-activating ligands in wild type, *BM-40-SPARC>shi[i]* and *>Tl[10B]* adipocytes assessed by real time RT-PCR. Error bars represent 95% confidence intervals. *rp49* expression was used for normalization.

This hyperconvoluted cell cortex is reminiscent of the basal labyrinth found in highly secretory cells, such as enterocytes and renal tubule epithelium. Fibrotic deposits in the absence of negative Toll regulator Cactus were similarly caused by PM overgrowth as well (*Figure 6*). In this instance, however, PM overgrowth resulted from increased membrane input from the secretory pathway, consistent with the ability of Toll to induce massive secretion of antimicrobial peptides. Supporting this, knock down of Rab1, required for ER-to-Golgi transport, was able to suppress Toll-induced PM overgrowth. Therefore, both recycling endosome-derived and ER-derived membrane inputs may contribute to total PM amount while constant endocytic membrane removal prevents PM overgrowth.

Sudden surges or shifts in membrane traffic are well known drivers of cell shape changes during development (*Lecuit and Pilot, 2003*). However, the extent to which maintenance of a stable cell shape is affected by the balance of membrane outputs and inputs to the PM has been less explored. Our experiments show that endocytosis is critical for preserving cortex morphology in *Drosophila* adipocytes and that preventing endocytosis can cause dramatic PM overgrowth. A factor that might determine high PM turnover in these cells, and thus a tendency to imbalances, is their elevated secretory activity. Apart from Collagen IV, fat body adipocytes secrete most other proteins present in the hemolymph, including the very abundant larval serum proteins (*Guedes Sde et al., 2003*; *Vierstraete et al., 2003*), which imposes a constant input of ER membrane to the PM. A need for control of PM amount, however, does not seem exclusive to adipocytes or highly secretory cells based on reported effects in other cell types. For instance, in Garland cells, a filtering cell type with high endocytic activity, *shibire* loss causes deep PM tubulation (*Kosaka and Ikeda, 1983*), whereas loss of *Rab11* flattens the PM (*Satoh et al., 2005*). In neurons, accumulation of membrane cisternae in

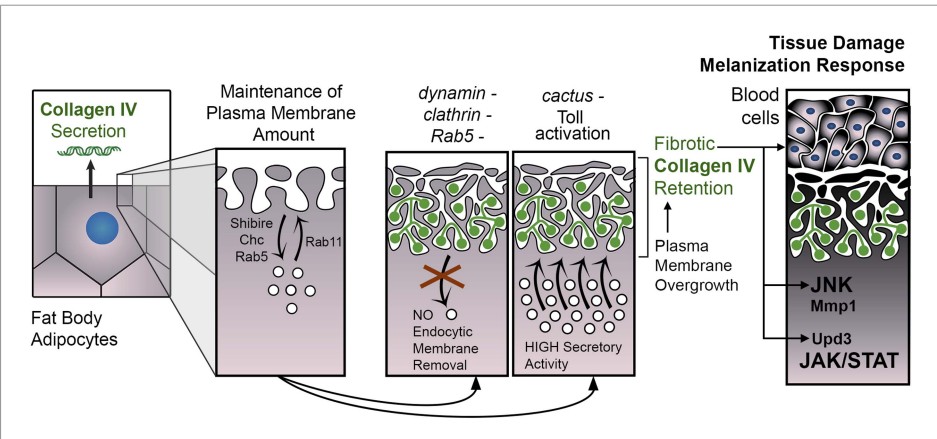

**Figure 8**. PM overgrowth leads to adipocyte fibrosis. Schematic representation summarizing the genesis of fibrotic deposits caused by PM excess and the ensuing reaction by the immune system. Defective endocytosis or excess secretion induced by Toll activity cause PM overgrowth in fat body adipocytes, which leads to hyperconvoluted PM morphology and pericellular trapping of Collagen IV and other extracellular matrix (ECM) proteins in the cell cortex. Fibrotic Collagen IV deposits trigger an immune response, as evidenced by tissue melanization and activation of the JAK/STAT and JNK pathways.

The following figure supplement is available for figure 8:

**Figure supplement 1**. Knock-down of *BM-40-SPARC* (*BM-40-SPARC>BM-40-SPARC[i]*) causes PM accumulation of Collagen IV without PM overgrowth.

synaptic boutons was observed upon Clathrin and Dynamin photoinactivation (*Heerssen et al., 2008*; *Kasprowicz et al., 2014*), similar to the PM excess we found in adipocytes. Finally, *Rab5* and other endocytic genes are required to maintain epithelial polarity (*Lu and Bilder, 2005*; *Zeigerer et al., 2012*), suggesting that constant endocytosis actively keeps apical and baso-lateral PMs segregated. These examples, together with our findings, point to a requirement for endocytosis and recycling in maintaining stable cell shape wider than currently appreciated.

In addition to the finding that membrane traffic controls adipocyte PM amount, our results have implications for the understanding of fibrotic deposits. We arrived at the study of membrane traffic in adipocytes through a Collagen IV accumulation phenotype. Collagen IV, however is not the only protein in these deposits (*Figure 5*). Also Perlecan, which we show originates in adipocytes as well (*Figure 4*), accumulates pericellularly. Except for Nidogen, a third basement membrane component, no other protein we tested accumulated, including Ferritin and Catalase, two of the most abundant proteins in the blood (*Handke et al., 2013*). Even though a more complete view of the aggregates must await a proteomic analysis, our data suggest that Collagen IV, while not the only protein present in these deposits, it is the most important one, as knock down of Collagen IV suppressed accumulation of Perlecan and Nidogen in *shi^i* cells. Because of its abundance and ability to self-interact, Collagen IV seems particularly prone in principle to nucleate aggregates in the conditions of limited diffusion provided by a hyperconvoluted PM. That Collagen IV self–interaction is critical in triggering pericellular aggregation is indeed supported by the fact that knock-down of *PH4α-EFB*, required for trimer formation, suppresses aggregate formation. In contrast, our data showing that PM surrounding aggregates can be stained equally well with both low (3000) and high (70,000) molecular weight fixable dextrans (*Figure 2—figure supplement 1*) suggests that the size of the accumulated protein does not play a major role in the formation of the deposits. This is also supported by the finding that Perlecan, a large 400 kDa protein, does not accumulate by itself, but as a result of Collagen IV accumulation (*Figure 5*).

Besides endocytic genes and *cactus*, our screening has found additional hits causing Collagen IV accumulation at the PM (*Supplementary file 1*). Loss of *BM-40-SPARC*, one of these hits, was previously known to cause Collagen IV deposits on adipocytes (*Pastor-Pareja and Xu, 2011*). Furthermore, similar to the fibrotic deposits in our study, other basement membrane components accumulate upon loss of *BM-40-SPARC* (*Shahab et al., 2015*). However, the PM is not overgrown in *BM-40-SPARC^i* adipocytes (*Figure 8—figure supplement 1*), suggesting that the way in which aggregates form in *BM-40-SPARC^i* adipocytes is different from the way they arise in endocytosis-defective and Toll-activated adipocytes. We have also tested Toll pathway activation by examining expression of Toll target Drosomycin (Drs-GFP) in *BM-40-SPARC^i* and all other PM accumulation hits, but found no Drosomycin-GFP induction in any of them. This suggests that, *cactus* aside, none of the hits causes Collagen IV deposits through ectopic Toll activation. Further characterization work, therefore, is needed to ascertain the particular mechanisms by which PM accumulation of Collagen IV occurs upon loss of these genes. Such work might offer additional insights for a better understanding of fibrosis.

## Adipocyte fibrosis

Pericellular trapping of Collagen IV, apart from preventing its incorporation to destination tissues, induces a potent response in adipocytes. Aspects of this response include melanization and activation of the JAK/STAT and JNK pathways (*Figure 7*). Concomitant activation of JNK and JAK/STAT signaling makes this response reminiscent of tissue damage responses in imaginal discs and gut tissues (*Pastor-Pareja et al., 2008*; *Buchon et al., 2009*). In particular, predominance of upd3 among induced JAK/STAT cytokines is shared with the intestinal response to tissue damage (*Buchon et al., 2009*). It has been suggested that basement membrane disruption could be a signal that intestinal cells sense when flies are fed tissue damage-inducing Dextran Sulfate Sodium (*Amcheslavsky et al., 2009*). Absence of basement membrane, although unable by itself to elicit a response (*Pastor-Pareja et al., 2008*), has been recently linked to melanization as a permissive factor (*Hauling et al., 2014*; *Kim and Choe, 2014*). In the case of adipocyte fibrosis, however, our data indicate that it is the fibrotic deposits that cause the response rather than absence of a basement membrane, since Collagen IV knock down in *shi^i* cells suppressed the response instead of enhancing it. Also in Toll[10B] adipocytes, Collagen IV reduction partially suppressed melanization, showing that, even though Toll

targets include melanization enzymes (*De Gregorio et al., 2002*), Collagen deposits contribute to Toll-induced melanization. An interesting question deserving of further investigation is how the fly immune system can react to these fibrotic deposits. One possibility is that fibrotic deposits are directly detected as altered self by the immune system, for instance through recognition of some specific molecular trait in them. A second possibility is that deposits somehow cause damage in adipocytes and this damage activates the response. In this latter case, the nature of the fibrosis-induced cell damage and the way that damage might be sensed are also worth deeper investigation.

Adipocyte fibrosis in humans and mouse models is correlated with obesity and adipose tissue dysfunction (*Sun et al., 2013*). Same as human adipocytes, *Drosophila* larval adipocytes are surrounded by a basement membrane containing Collagen IV (*Chun, 2012*); also similar, human adipocytes display complex PM topology, with abundant caveolae forming higher-level rosettes (*Fan et al., 1983*). Our study, importantly, provides a model to study fibrosis in this tissue. Fibrosis, affecting adipocytes or otherwise, is thought to result in all cases from inflammatory stimulation of excess ECM deposition. Our results show that accumulation of trimeric Collagen IV can be triggered by Toll immune activation and that deposits in turn further stimulate other arms of the fly immune system different from the Toll-mediated immune response. This raises the possibility that fibrotic accumulations, once initiated, could be self-sustained, with inflammation and fibrosis feeding each other. Interestingly, the response to fibrosis includes induction of Collagen-degrading enzyme Matrix metalloprotease 1, suggesting that the response is aimed in part at clearing the aggregates. Further research into how innate immunity is activated by these deposits may uncover therapeutically relevant mechanisms by which adipocytes and other tissues react to, and deal with, excessive matrix deposition.

## Materials and methods

### *Drosophila* strains and culture

Standard fly husbandry techniques and genetic methodologies, including balancers and dominant genetic markers, were used to assess segregation of mutations and transgenes in the progeny of crosses, construct intermediate fly lines and obtain flies of the required genotypes for each experiment (*Roote and Prokop, 2013*). Flies were maintained at 25°C unless otherwise stated. In the initial screening and most experiments afterwards, the GAL4-UAS binary expression system (*Brand and Perrimon, 1993*) was used to drive expression of UAS transgenes in fat body adipocytes under temporal and spatial control of transgenic GAL4 drivers *BM-40-SPARC-GAL4* (*Venken et al., 2011*) (a gift from Hugo Bellen), *Cg-GAL4* (*Asha et al., 2003*) (BL7011), *r4-GAL4* (*Lee and Park, 2004*) (a gift from Pierre Leopold) and *ppl-GAL4* (*Colombani et al., 2003*) (a gift from Herve Agaisse). In flies bearing both types of transgenes (a GAL4 driver and a UAS responder), expression of the UAS-transgene is induced by the yeast transcription factor GAL4 according to the pattern of expression of GAL4 specific to the driver transgene. The strength of these drivers is *BM-40-SPARC-GAL4 > Cg-GAL4 > r4-GAL4 > ppl-GAL4*. While *BM-40-SPARC-GAL4* and *Cg-GAL4* are expressed in blood cells, *r4-GAL4* and *ppl-GAL4* are not. UAS-Dcr2 was included in the screening strain for the purpose of enhancing RNAi-mediated knock-down by long dsRNA hairpins (*Dietzl et al., 2007*). Dcr2 expression in adipocytes showed no visible effect by itself on Collagen IV localization (*Figure 1—figure supplement 2*). For generation of flip-out clones (*Ito et al., 1997*) (*Figure 2F,G*), vials containing L2 larvae were heat-shocked at 37°C for 7 min. For lipid depletion experiments, lipids were extracted from medium by mixing ingredients with chloroform for 2 days, allowing chloroform to evaporate for at least two more days before food preparation (*Palm et al., 2012*). Genotypes of animals in all experiments are detailed in *Supplementary file 2*. Origin of mutants and transgenes used can be found in *Supplementary file 3*. The following strains were used:

Canton-S
$w^{1118}$
*y w; vkg$^{G454}$/CyO*
*w; vkg$^{G454}$ UAS-myr-RFP; BM-40-SPARC-GAL4 UAS-Dcr2/SM6a-TM6B*
*w; vkg$^{G454}$/CyO; BM-40-SPARC-GAL4 UAS-myr.RFP/TM6B*
*w; Cg-GAL4 (II)*
*w; Cg-GAL4 UAS-myr-RFP (II)*

*w; ppl-GAL4* (II)
*w; ppl-GAL4 UAS-myr.RFP vkg^{G454}/CyO*
*w; ppl-GAL4 UAS-myr.RFP vkg^{G454}/CyO; UAS-Dcr2*
*w; r4-GAL4* (III)
*w; UAS-myr.RFP vkg^{G454}/CyO; r4-GAL4*
*y sc v; UAS-shi.RNAi^{TRiP.JF03133}*
*y sc v; UAS-shi.RNAi^{TRiP.HMS00154}*
*w; UAS-shi.RNAi^{NIG.18102R-1}*
*w; UAS-shi.K44A.3-7* (II)
*w shi^1*
*shi^2*
*y w hs-Flp1.22; vkg^{G454} act-y+-GAL4 UAS-myr.RFP/CyO*
*y w hs-Flp1.22; act-y+-GAL4 UAS-GFP*
*w; UAS-Cg25C.GFP.2.1* (II)
*w; UAS-Cg25C.RFP.2.1* (II)
*w; UAS-hTfR.GFP* (III)
*y sc v; UAS-Rab11.RNAi^{TRiP.JF02812}*
*w UAS-Rab11.dsRNA.WIZ*
*y sc v; UAS-Rab5.RNAi^{TRiP.HMC03420}*
*y sc v; UAS-Rab5.RNAi^{TRiP.HMS00147}*
*w; UAS-Rab5.S43N* (II) (BL42703)
*w; UAS-GFP.Rab5* (III) (BL43336)
*y v sc; UAS-Chc.RNAi^{TRIP.JF02681}*
*y v sc; UAS-Chc.RNAi^{TRIP.HMS01222}*
*w; UAS-Chc.DN* (III) (BL26874)
*y v sc; UAS-RN-tre.RNAi^{TRiP.JF03085}* (III)
*y sc v; UAS-AP-2α.RNAi^{TRiP.HMS00653}* (III)
*y sc v; UAS-AP-2μ.RNAi^{TRiP.JF0287}* (III)
*y sc v; UAS-Hrs.RNAi^{TRiP.JF02860}* (III)
*y sc v; UAS-Par-1.RNAi^{TRiP.GL00253}* (III)
*w trol^{CPTI-002049}*
*y sc v; UAS-EGFP.shRNA.3* (II) (BL41559)
*y sc v; UAS-EGFP.shRNA.3* (III) (BL41560)
*y sc v; UAS-EGFP.shRNA.4* (III) (BL41553)
*y sc v; UAS-EGFP.shRNA* (II) (BL35782) (knocks down GFP and YFP efficiently same as three above constructs; unlike those, it causes unspecific cell death in imaginal discs)
*w; UAS-vkg.RNAi^{NIG.16858R-3}* (III)
*w; UAS-vkg.RNAi^{VDRC.v16986}* (III)
*w; UAS-vkg.RNAi^{VDRC.v106812}* (II)
*w; UAS-Cg25C.RNAi^{VDRC.v28369}* (III)
*w; UAS-vkg.RNAi^{NIG.16858R-3} UAS- Cg25C.RNAi^{VDRC.v28369}/TM6B*
*w; UAS-PH4αEFB.RNAi^{VDRC.v2464}* (III)
*w; UAS-sec23.RNAi^{VDRC.v24552}* (III)
*w; 26-29-p^{CA06735}* (III)
*w; Fer1HCH^{G188}* (III)
*y sc v; UAS-cact.RNAi^{TRiP.GL00627}* (II)
*y sc v; UAS-cact.RNAi^{TRiP.HMS00084}* (III)
*cact^4/CyO, act-GFP*
*Df(2L)r10, cn^1/CyO*
*w; UAS-Tl^{10B}* (II)
*y sc v; UAS-Dif.RNAi ^{TRiP.HM05191}* (III)
*y w Drs-GFP.JM804*
*y sc v; UAS-Rab1.RNAi^{TRiP.JF02609}* (III)
*w; puc^{G462}/TM6B*
*w; 10XSTAT-GFP* (II)

## UAS-Cg25C-GFP and UAS-Cg25C-RFP transgenic lines

To obtain UAS-Cg25C-GFP and UAS-Cg25C-RFP lines, the open reading frame (ORF) of Cg25C was cloned into vectors pTWG and pTWR (Drosophila Carnegie Vector collection) using Gateway recombination (Life Technologies, Carlsbad, California). To do this, we PCR-amplified the Cg25C ORF from cDNA clone RE33133 (Drosophila Genomics Resource center, Bloomington) using primers Cg25C-NF: 5′-GGGG ACA AGT TTG TAC AAA AAA GCA GGC TTC ATG TTG CCC TTC TGG AAG CG-3′ and Cg25C-CF: 5′-GGG GAC CAC TTT GTA CAA GAA AGC TGG GTC CGA GGA GTT CTT CAT GCA CA-3′, thus adding att sites at the 5′ and 3′ termini of the ORF for subsequent Gateway cloning. The product of this reaction was purified by gel extraction with AxyPrep DNA Gel Extraction Kit (Axygen, Union City, California) and recombined into vector pDONR221 (Life Technologies) with Gateway BP Clonase Enzyme Mix (Life Technologies). The resulting plasmid (Entry clone), was transformed into *Escherichia coli* Trans5α competent cells, miniprepped and recombined with destination vectors pTWG (C-terminal GFP) and pTWR (C-terminal RFP) using Gateway LR Clonase Enzyme Mix (Life Technologies). Transgenic lines were obtained through standard P-element transgenesis (*Spradling and Rubin, 1982*).

## Immunohistochemistry

For generation of anti-Cg25C antibody, rabbits were immunized with epitope SVKHYNRNEPKFPIDDSY by AbMax Biotechnology (Beijing, China). The following antibodies and dyes were used: rabbit anti-Cg25C (1:1,000, this study), mouse anti-rat Dynamin (1:250, BD BioScience, Franklin Lakes, New Jersey), rabbit anti-Ndg (*Wolfstetter et al., 2009*) (1:2000), rabbit anti-Pcan (*Friedrich et al., 2000*) (1:2000), rabbit anti-LanB1 (1:500, Abcam, Cambridge, UK), mouse anti-Mmp1 (*Page-McCaw et al., 2003*) (1: 200, DSHB, University of Iowa), mouse anti-Dorsal (*Whalen and Steward, 1993*) (1:100, DSHB), rabbit anti-GFP (1:500, EASYBIO, Seoul, South Korea), anti-mouse IgG conjugated to Alexa-488, Alexa-555 or Alexa-633 (1:200, Life Technologies), Texas-Red phalloidin (1:100, Life Technologies) and BODIPY 493/503 (1:500, Life Technologies). Antibody stainings of whole fat body were performed following standard procedures used for imaginal discs. Briefly, larvae were pre-dissected in PBS by turning inside out with fine tip forceps after removing their posterior end. Carcasses with fat body still attached were fixed in PBS containing 4% PFA, washed in PBS (3 × 10 min), blocked in PBT-BSA (PBS containing 0.1% Triton X-100 detergent, 1% BSA and 250 mM NaCl), incubated overnight with primary antibody in PBT-BSA, washed in PBT-BSA (3 × 20 min), incubated for 2 hr with secondary antibody in PBT-BSA and washed in PBT-BSA (3 × 20 min) and PBS (3 × 10 min). Fat body tissues were finally dissected on a slide with a drop of DAPI-Vectashield (Vector Labs, Burlingame, California) and mounted in this same medium. FM4-64 staining of PM was performed by incubating fat body in ice-cold PBS containing 5 mg/ml fixable FM4-64 (Life Technologies) for 1 min before fixing for 20 min in ice-cold PBS containing 4% PFA. Labeling with fixable 3,000 and 70,000 MW Texas-Red-coupled dextran (0.1 mg/ml, Life Technologies) was performed by fixing fat body in 4% PFA for 5 min followed by three quick washes, incubation with dextran in PBS, three quick washes and fixation again for 20 min. Results were the same when the first fixation was omitted.

## Imaging

The screening was conducted in a Leica MZ10F stereoscope (Wetzlar, Germany). For confocal microscopy (Zeiss LSM780 microscope, Oberkochen, Germany), tissues were mounted in DAPI-Vectashield (Vector Labs). Vkg-GFP intensity in basement membranes was quantified with ImageJ software as previously described (*Pastor-Pareja and Xu, 2011*). PM depth and sinuosity measurements were performed in electron micrographs as indicated in *Figure 3—figure supplement 2*. For measurements of both PM depth and sinuosity, images were analyzed with ImageJ using the segmented line tool to calculate distances. Graphs were drawn and statistical analysis performed with GraphPad Prism. For sinuosity measurements, the distance separating the points where the PM intersects the edges of the micrograph was calculated, as well as the length of the PM joining those two points. Islands of cytoplasm with no connection to the cell body were excluded from sinuosity analysis, but not for PM depth measurement. All fat body images in the manuscript correspond to mid-to-late third larval instar fat body dissected prior to the wandering larva stage, except for the first and second larval instar images in *Figure 3—figure supplement 1A* and sec23[i] second instar larvae in *Figure 5—figure supplement 1B*. Fibrotic aggregates can be observed at all times during the third larval instar, including the extremes of early third instar and white prepupa.

## Transmission electron microscopy

Ultrathin sections were obtained following standard procedures and imaged in a Hitachi H-7650B microscope (Tokyo, Japan). Briefly, larvae were turned inside-out and fixed in 2.5% gluteraldehyde, leaving fat body attached to carcasses to facilitate handling. Once fixed, fat body was postfixed in 1% osmium tetroxide before embedding. Sections were stained in 2% uranyl acetate/lead citrate. Fat body from at least three different specimens per genotype was imaged.

## Bacterial infection

Gram+ bacterium *M. luteus* (ATCC 4698) was cultured in LB medium at 30°C. Larvae were soaked in concentrated culture and pricked with a tungsten needle to allow infection.

## Real time RT-PCR

RNA was isolated using TransZol Up (Transgen, Beijing, China). cDNA was synthesized from 1 µg of RNA with the PrimeScriptTM RT-PCR Kit (Takara, Kyoto, Japan). Analysis was performed in a CFX96 Touch system (Bio-Rad, Hercules, California) using SYBR Green Fast kit (Applied Biosystems, Waltham, Massachusetts). *rp49* expression was used for normalization. Three experiments per genotype were averaged. A biological replicate was performed with same results. Primers used were:

upd-For: 5′ TCCACACGCACAACTACAAGTTC 3′;
upd-Rev: 5′ CCAGCGCTTTAGGGCAATC 3′;
upd2-For: 5′ AGTGCGGTGAAGCTAAAGACTTG 3′;
upd2-Rev: 5′ GCCCGTCCCAGATATGAGAA 3′;
upd3-For: 5′ TGCCCCGTCTGAATCTCACT 3′;
upd3-Rev: 5′ GTGAAGGCGCCCACGTAA 3′;
rp49-For: 5′ GGCCCAAGATCGTGAAGAAG 3′;
rp49-Rev: 5′ ATTTGTGCGACAGCTTAGCATATC 3′.

## Western blots of hemolymph

Hemolymph from third instar larvae was collected by turning larvae inside-out with fine-tip forceps while immersed in 100 µl of PBS in a glass depression well. PBS contained 0.1 mg/ml of phenylthiourea to avoid melanization and 1 mg/ml of protease inhibitor. 10 µl of 2-Mercaptoethanol-reduced sample were loaded per genotype into a 4–15% Mini-PROTEAN TGX gel (Bio-Rad). The amount of sample loaded in each experiment given in figure legends as larva-equivalents, depending on the number of larvae collected. Precision Plus Protein Dual Color Standards (Bio-Rad) was used as a molecular weight marker. Gels were blotted with rabbit anti-GFP antibody (1:5000, EASYBIO) or rabbit anti-Cg25C (1:5000, this study) and revealed with anti-rabbit-HRP (1:10,000, Abmart) and an ECL Plus kit (Pierce, Rockford, Illinois).

## Acknowledgements

We thank Hugo Bellen (*BM-40-SPARC-GAL4*), Pierre Leopold (*r4-GAL4*), Herve Agaisse (*ppl-GAL4*), Fujian Zhang (UAS-secr-GFP) and Stefan Baumgartner (anti-Trol and anti-Ndg) for sharing fly strains and antibodies; the Bloomington Stock Center, National Institute of Genetics (Kyoto) and Vienna Drosophila RNAi Center for providing strains; the Drosophila Genomics Research Center (Indiana) for Cg25C cDNA and the Drosophila Carnegie Vector collection, and the Tsinghua Electron Microscopy facility. Xingxin Liao, Huina Jia, Jiatao Li, Xiaoxue Chen and Yuxi Li assisted in screening and some experiments. Special thanks are due to Tian Xu, in whose laboratory planning for the screening took place, and to Tatsushi Igaki and Helen Rankin for comments on the manuscript.

## Additional information

### Funding

| Funder | Grant reference | Author |
| --- | --- | --- |
| National Natural Science Foundation of China (NSFC) | General program grant, 31371459 | José Carlos Pastor-Pareja |

| Funder | Grant reference | Author |
| --- | --- | --- |
| State Administration of Foreign Experts Affairs | 1000 Young Talents award | José Carlos Pastor-Pareja |
| Tsinghua University | Tsinghua Initiative Program, 20131089281 | José Carlos Pastor-Pareja |

The funders had no role in study design, data collection and interpretation, or the decision to submit the work for publication.

## Author contributions

YZ, MW, ML, JCP-P, Conception and design, Acquisition of data, Analysis and interpretation of data, Drafting or revising the article; HK, Acquisition of data, Analysis and interpretation of data, Drafting or revising the article; SM, Acquisition of data, Drafting or revising the article; L-PL, J-QN, Conception and design, Drafting or revising the article, Contributed unpublished essential data or reagents

## Author ORCIDs

José Carlos Pastor-Pareja, ⓘ http://orcid.org/0000-0002-3823-4473

## Additional files

### Supplementary files

• Supplementary file 1. Hits in RNAi screening causing plasma membrane accumulation of Collagen IV in adipocytes.

• Supplementary file 2. Detailed genotypes.

• Supplementary file 3. Origin of antibodies, mutants and transgenes used in this study.

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
