## [Decision Letter]

Thank you for sending your work entitled “Plasma Membrane Overgrowth Causes Fibrotic Collagen Retention and Immune Activation in *Drosophila* Adipocytes” for consideration at *eLife*. Your article has been favorably evaluated by Tadatsugu Taniguchi (Senior editor), a Reviewing editor, and three reviewers.

1) The authors argue that collagen deposition underlies fibrosis. To support this claim, they show that knocking down PH4αEFB rescues the dynamin mutant phenotype. It is important to perform the same experiment in the context of Toll pathway activation to demonstrate that the same process is at work.

2) Additional experiments are required to clarify whether collagen is extracellular or intracellular. This is a source of confusion in the paper. More specifically: Is there a way to show that their plasma membrane inclusions that they see surrounding adipocytes are actually open to the extracellular space? The FM dye was not really convincing and the paper really hinges on the idea that these inclusions are not internal vesicles. Could the authors whole mount stain the fat body for Collagen IV without permeabilizing? If the antibody can still get access to the Collagen then it would be good evidence that it is really extracellular. It may be that the authors actually performed this experiment (Figure 1) as they do show antibody staining of fat body for Collagen IV. However, they do not tell us if these were sections or whole mounts, or whether there was any permeabilization.

Related to the comment above, is the Collagen IV inclusions related to the size of the collagen? If the authors bathe the fat body in different size dextrans would they be able to highlight these plasma membrane inclusions and show that there is a threshold size that is incapable of getting into these pockets? This may be an additional way to show that they really are extracellular. To ensure that the dextran is not being taken up by endocytosis they may need to do this experiment on fixed tissue.

3) Confirm that the secretion of other proteins is not affected in the condition tested by looking to the presence of secreted proteins in the hemolymph. More specifically, it would be good for the authors to show that in their various perturbations, they get reduced levels of Collagen IV in the hemolymph (or are capable of rescuing Collagen IV levels in the hemolymph). Presumably most of the Collagen IV is trapped around the adipocytes at the expense of soluble Collagen when they disturb trafficking? The authors also show some negative controls regarding proteins that are not trapped in the PM inclusions (secr-GFP) in the dissected fat body. But they don't show that this secr-GFP is actually secreted in their system (it would be good to see a positive control that this is actually expressed). Could the authors do this in vivo and image the hemolymph within larvae and look at GFP levels? Or, similar to the comment above, look for GFP levels in the hemolymph by western blot? The same should be said for the other control proteins that they examine (Ferritin, 26-29p and catalase).

4) The authors artificially decrease PM amount by depleting lipids. This is a surprising result that should be substantiated. They show that this rescues PM defects by looking at plasma membrane depth by confocal. It was unclear why they didn't also perform EM and also use their other measures of PM surface area as in their other perturbations. This should be performed for consistency.

5) The Methods section needs to be expanded to make it more understandable to readers not familiar with the genetic crosses.

6) About the other hits of the screen: they should be briefly discussed or eliminated from the text.

---

## [Author Response]

*1) The authors argue that collagen deposition underlies fibrosis. To support this claim, they show that knocking down PH4αEFB rescues the dynamin mutant phenotype. It is important to perform the same experiment in the context of Toll pathway activation to demonstrate that the same process is at work*.

We wish to thank the editor and reviewers for the effort they have put into reviewing and improving our manuscript. We have performed the suggested rescue experiments and now show with confocal images that, same as with the dynamin mutant phenotype, *PH4αEFB* knock-down suppresses pericellular accumulation of Collagen IV in both *cact*^*i*^ and Toll^10B^ adipocytes. This is now shown in Figure 6 and mentioned in the text (subsection “Fibrotic deposits and plasma membrane overgrowth upon Toll activation”):

“Moreover, pericellular Collagen IV deposits were extracellular (antibody staining without permeabilization, Figure 6—figure supplement 1), disappeared upon *PH4αEFB* knock-down (Figure 6) and did not contain secr-GFP (Figure 6), but contained Perlecan (Figure 6), indicating accumulation of fibrotic material, same as in endocytosis-defective cells.”

*2) Additional experiments are required to clarify whether collagen is extracellular or intracellular. This is a source of confusion in the paper. More specifically: Is there a way to show that their plasma membrane inclusions that they see surrounding adipocytes are actually open to the extracellular space? The FM dye was not really convincing and the paper really hinges on the idea that these inclusions are not internal vesicles. Could the authors whole mount stain the fat body for Collagen IV without permeabilizing? If the antibody* can *still get access to the Collagen then it would be good evidence that it is really extracellular. It may be that the authors actually performed this experiment (*Figure 1*) as they do show antibody staining of fat body for Collagen IV. However, they do not tell us if these were sections or whole mounts, or whether there was any permeabilization*.

*Related to the comment above, is the Collagen IV inclusions related to the size of the collagen? If the authors bathe the fat body in different size dextrans would they be able to highlight these plasma membrane inclusions and show that there is a threshold size that is incapable of getting into these pockets? This may be an additional way to show that they really are extracellular. To ensure that the dextran is not being taken up by endocytosis they may need to do this experiment on fixed tissue*.

We have performed additional experiments to demonstrate that Collagen IV accumulating in endocytosis-defective and Toll-activated adipocytes is in fact extracellular. We have done this in the two ways suggested by the referees.

First, with antibody stainings performed in absence of permeabilization, to detect extracellular Vkg-GFP (anti-GFP) and extracellular Cg25C (anti-Cg25C). Without permeabilization (no Triton X detergent treatment at any step), both Vkg and Cg25C are still detected at their pericellular location. As a control, we use intracellular retention of Collagen IV in *Tango1*^*i*^ adipocytes to show that intracellular Collagen cannot be detected when permeabilization is omitted. The new data are now shown in Figure 2 for *shi*^*i*^ and in Figure 6—figure supplement 1 for *cact*^*i*^ and Toll^10B^. It must be noted that previous stainings were done as whole mounts with permeabilization, which we now indicate in the Results section:

“To confirm that the Collagen IV accumulations in *shi*^*i*^ adipocytes were extracellular, we performed antibody stainings […] indicating that the accumulations were indeed extracellular.”

The second way, also as suggested, is using fluorescently-labeled dextran in fixed tissue. We now show results using low MW (4000) and high MW (70,000) fixable dextrans in Figure 2 and Figure 2—figure supplement 1. In both cases membranes surrounding aggregates are stained by the dextran particles after incubation for 2 min. These results further support that Collagen IV aggregates are surrounded by PM and thus extracellular (please see the Results section):

“Labeling of membrane around the accumulations was similarly observed when *shi*^*i*^ adipocytes were fixed and stained with fluorescently-labeled fixable dextrans of molecular weights 70,000 (Figure 2) and 3000 (Figure 2—figure supplement 1).”

Our experiments with dextrans also suggest that the size of the particles is not determinant for the accumulation. This is also supported by the fact that Perlecan, a 400,000 MW protein, is definitely not trapped by virtue of its size, but as a result of Collagen IV retention (Figure 5). We therefore give now more weight in the Discussion to the idea that the ability of Collagen to self-interact is the key factor determining aggregation and discourage the interpretation that Collagen IV protein size might play a major role:

“That Collagen IV self–interaction is critical in triggering pericellular aggregation […] but as a result of Collagen IV retention (Figure 5).”

We also tried using non-fixable fluorescein-coupled dextrans, for which a wider range of particle sizes is available, to try to define better a size threshold for accumulation but could not get any conclusive results, as those dextrans did not associate with membranes, neither in live nor in fixed tissues.

A third, additional way in which we are now supporting that Collagen IV is extracellular in endocytosis-defective and Toll-activated cells is by pointing out examples in electron micrographs where a connection to the extracellular space of the cavity where the aggregate(s) are located can be clearly discerned. For *shi*^*i*^ this is now shown in Figure 2—figure supplement 1. For *cact*^*i*^ and Toll^10B^ we now point out clear connections to the extracellular space in the images in Figure 6 and Figure 6—figure supplement 1.

*3) Confirm that the secretion of other proteins is not affected in the condition tested by looking to the presence of secreted proteins in the hemolymph. More specifically, it would be good for the authors to show that in their various perturbations, they get reduced levels of Collagen IV in the hemolymph (or are capable of rescuing Collagen IV levels in the hemolymph). Presumably most of the Collagen IV is trapped around the adipocytes at the expense of soluble Collagen when they disturb trafficking*?

That the amount of soluble Collagen IV is reduced as a result of pericellular retention is supported by data showing reduced deposition in basement membranes and VNC organ deformation (Figure 1 and Figure 6). To further support this, we have performed, as suggested, Western blot analysis of hemolymph in wild type, *shi*^*i*^, *cact*^*i*^ and Toll^10B^ larvae, which confirms reduced presence of Collagen in the hemolymph, especially in *shi*^*i*^. These data are now shown in Figure 1—figure supplement 1.

*The authors also show some negative controls regarding proteins that are not trapped in the PM inclusions (secr-GFP) in the dissected fat body. But they don't show that this secr-GFP is actually secreted in their system (it would be good to see a positive control that this is actually expressed). Could the authors do this* in vivo *and image the hemolymph within larvae and look at GFP levels? Or, similar to the comment above, look for GFP levels in the hemolymph by western blot*?

To confirm that secr-GFP (a signal peptide coupled to GFP) is actually secreted to the hemolymph when expressed in the fat body we have performed Western blot analysis of hemolymph with anti-GFP. A clear band of the expected size can be detected indeed (Figure 5—figure supplement 1). Moreover, as further proof that GFP is secreted when secr-GFP is expressed in the fat body, we now show that GFP can be detected in pericardial cells lining the larval heart, known to act as hemolymph-filtering nephrocytes (Figure 5—figure supplement 1). These new data, together with the intracellular retention of secr-GFP upon *Rab1* knock-down shown before (Figure 5), confirm that secr-GFP is secreted to the hemolymph. Although we cannot rule out the presence of other non-ECM proteins in the aggregates, we believe our results unmistakably show that not all secreted proteins will be trapped in the aggregates.

*The same should be said for the other control proteins that they examine (Ferritin, 26-29p and catalase)*.

We have also conducted Western blot analysis of hemolymph with anti-GFP for Fer1HCH-GFP and Drs-GFP, also obtaining clear bands of the expected size (Figure 5—figure supplement 1). We can confirm as well presence of 26-29-protease-GFP in the hemolymph, the band pattern suggesting the full length protein is processed to some extent. As for Catalase, we could not reliably detect presence of Catalase-YFP in the hemolymph with anti-GFP (not shown) and so have taken this protein from our list of controls as a precaution.

*4) The authors artificially decrease PM amount by depleting lipids. This is a surprising result that should be substantiated. They show that this rescues PM defects by looking at plasma membrane depth by confocal. It was unclear why they didn't also perform EM and also use their other measures of PM surface area as in their other perturbations. This should be performed for consistency*.

For the purpose of consistency, and to further support PM flattening in lipid-depleted adipocytes, we have documented their PM topology with electron micrographs (Figure 3—figure supplement 1) and the corresponding quantitative analysis, included in the graphs in Figure 3.

*5) The Methods section needs to be expanded to make it more understandable to readers not familiar with the genetic crosses*.

We have expanded this section to try to explain to the non-Drosophilists the logic of the genetic methods we employed. In particular, since most experiments involve GAL4-driven transgene expression, we now explain the basics of the GAL4-UAS system and give more details of the GAL4 drivers we have used:

“Standard fly husbandry techniques and genetic methodologies, including balancers and dominant genetic markers, were used to assess segregation of mutations […] Genotypes of animals in all experiments are detailed in [Supplementary-material SD2-data].”

*6) About the other hits of the screen: they should be briefly discussed or eliminated from the text*.

Since some of the later questions by the referees deal with the content of that list (e.g. *BM-40-SPARC*, chromatin hits), we feel inclined to keep these data in the manuscript. We now devote a paragraph in the Discussion (“Besides endocytic genes and *cactus*, our screening has found additional hits causing […] additional insights for a better understanding of fibrosis”) to point out that:

A) *BM-40-SPARC*^*i*^ causes PM retention of Collagen IV (58; 68), but no PM overgrowth (data now shown in Figure 8—figure supplement 1).

B) No other hit in the screening apart from *cactus* seems to activate Toll signaling, assayed by Drosomycin-GFP induction.

C) Genes potentially related to fibrosis may be contained in that list.